# Collaborative Filtering with Graph Information: Consistency and Scalable Methods

**Nikhil Rao**    **Hsiang-Fu Yu**    **Pradeep Ravikumar**    **Inderjit S. Dhillon**

{nikhilr, rofuyu, paradeepr, inderjit}@cs.utexas.edu
Department of Computer Science
University of Texas at Austin

## Abstract

Low rank matrix completion plays a fundamental role in collaborative filtering applications, the key idea being that the variables lie in a smaller subspace than the ambient space. Often, additional information about the variables is known, and it is reasonable to assume that incorporating this information will lead to better predictions. We tackle the problem of matrix completion when pairwise relationships among variables are known, via a graph. We formulate and derive a highly efficient, conjugate gradient based alternating minimization scheme that solves optimizations with over 55 million observations up to 2 orders of magnitude faster than state-of-the-art (stochastic) gradient-descent based methods. On the theoretical front, we show that such methods generalize weighted nuclear norm formulations, and derive statistical consistency guarantees. We validate our results on both real and synthetic datasets.

## 1 Introduction

Low rank matrix completion approaches are among the most widely used collaborative filtering methods, where a partially observed matrix is available to the practitioner, who needs to impute the missing entries. Specifically, suppose there exists a ratings matrix $Y \in \mathbb{R}^{m \times n}$, and we only observe a subset of the entries $Y_{ij}, \forall (i,j) \in \Omega, |\Omega| = N \ll mn$. The goal is to estimate $Y_{i,j}, \forall (i,j) \notin \Omega$. To this end, one typically looks to solve one of the following (equivalent) programs:

$$\hat{Z} = \arg \min_{Z} \frac{1}{2} \|\mathcal{P}_\Omega (Y - Z)\|_F^2 + \lambda_z \|Z\|_* \tag{1}$$

$$\hat{W}, \hat{H} = \arg \min_{W,H} \frac{1}{2} \|\mathcal{P}_\Omega (Y - WH^T)\|_F^2 + \frac{\lambda_w}{2} \|W\|_F^2 + \frac{\lambda_h}{2} \|H\|_F^2 \tag{2}$$

where the nuclear norm $\|Z\|_*$, given by the sum of singular values, is a tight convex relaxation of the non convex rank penalty, and is equivalent to the regularizer in (2). $\mathcal{P}_\Omega (\cdot)$ is the projection operator that only retains those entries of the matrix that lie in the set $\Omega$.

In many cases however, one not only has the partially observed ratings matrix, but also has access to additional information about the relationships between the variables involved. For example, one might have access to a social network of users. Similarly, one might have access to attributes of items, movies, etc. The nature of the attributes can be fairly arbitrary, but it is reasonable to assume that "similar" users/items share "similar" attributes. A natural question to ask then, is if one can take advantage of this additional information to make better predictions. In this paper, we assume that the row and column variables lie on graphs. The graphs may naturally be part of the data (social networks, product co-purchasing graphs) or they can be constructed from available features. The idea then is to incorporate this additional structural information into the matrix completion setting.

We not only require the resulting optimization program to enforce additional constraints on $Z$, but we also require it to admit efficient optimization algorithms. We show in the sections that follow that this in fact is indeed the case. We also perform a theoretical analysis of our problem when the observed entries of $Y$ are corrupted by additive white Gaussian noise. To summarize, the contributions of our paper are as follows:

- We provide a *scalable* algorithm for matrix completion graph with structural information. Our method relies on efficient Hessian-vector multiplication schemes, and is orders of magnitude faster than (stochastic) gradient descent based approaches.
- We make connections with other structured matrix factorization frameworks. Notably, we show that our method generalizes the weighted nuclear norm [21], and methods based on Gaussian generative models [27].
- We derive consistency guarantees for graph regularized matrix completion, and empirically show that our bound is smaller than that of traditional matrix completion, where graph information is ignored.
- We empirically validate our claims, and show that our method achieves comparable error rates to other methods, while being significantly more scalable.

**Related Work and Key Differences**

For convex methods for matrix factorization, Haeffele et al. [9] provided a framework to use regularizers with norms other than the Euclidean norm in (2). Abernethy et al. [1] considered a kernel based embedding of the data, and showed that the resulting problem can be expressed as a norm minimization scheme. Srebro and Salakhutdinov [21] introduced a weighted nuclear norm, and showed that the method enjoys superior performance as compared to standard matrix completion under a non-uniform sampling scheme. We show that the graph based framework considered in this paper is in fact a generalization of the weighted nuclear norm problem, with non-diagonal weight matrices.

In the context of matrix factorization with graph structural information, [5] considered a graph regularized nonnegative matrix factorization framework and proposed a gradient descent based method to solve the problem. In the context of recommendation systems in social networks, Ma et al. [14] modeled the weight of a graph edge[1] explicitly in a re-weighted regularization framework. Li and Yeung [13] considered a similar setting to ours, but a key point of difference between all the aforementioned methods and our paper is that we consider the partially observed ratings case. There are some works developing algorithms for the situation with partially observations [12, 26, 27]; however, none of them provides statistical guarantees. Weighted norm minimization has been considered before ([16, 21]) in the context of low rank matrix completion. The thrust of these methods has been to show that despite suboptimal conditions (correlated data, non-uniform sampling), the sample complexity does not change. None of these methods use graph information. We are interested in a complementary question: *Given variables conforming to graph information, can we obtain better guarantees under uniform sampling to those achieved by traditional methods?*

## 2 Graph-Structured Matrix Factorization

Assume that the "true" target matrix can be factorized as $Z^\star = W^\star (H^\star)^T$, and there exist a graph $(V^w, E^w)$ whose adjacency matrix encodes the relationships between the $m$ rows of $W^\star$ and a graph $(V^h, E^h)$ for $n$ rows of $H^\star$. In particular, two rows (or columns) connected by an edge in the graph are "close" to each other in the Euclidean distance. In the context of graph-based embedding, [3, 4] proposed a smoothing term of the form

$$\frac{1}{2} \sum_{i,j} E_{ij}^w (\boldsymbol{w}_i - \boldsymbol{w}_j)^2 = \operatorname{tr}(W^T \operatorname{\mathbf{Lap}}(E^w) W) \tag{3}$$

where $\operatorname{\mathbf{Lap}}(E^w) := D^w - E^w$ is the graph Laplacian for $(V^w, E^w)$, where $D^w$ is the diagonal matrix with $D_{ii}^w = \sum_{j \sim i} E_{ij}^w$. Adding (3) into the minimization problem (2) encourages solutions where $\boldsymbol{w}_i \approx \boldsymbol{w}_j$ when $E_{ij}^w$ is large. A similar argument holds for $H^\star$ and the associated graph Laplacian $\operatorname{\mathbf{Lap}}(E^h)$.

We would thus not only want the target matrix to be low rank, but also want the variables $W, H$ to be faithful to the underlying graph structure. To this end, we consider the following problem:

$$\min_{W,H} \frac{1}{2}\|\mathcal{P}_\Omega\left(Y - WH^T\right)\|_F^2 + \frac{\lambda_L}{2}\{\text{tr}(W^T\,\textbf{Lap}(E^w)W) + \text{tr}(H^T\,\textbf{Lap}(E^h)H)\} + \quad (4)$$

$$\frac{\lambda_w}{2}\|W\|_F^2 + \frac{\lambda_h}{2}\|H\|_F^2$$

$$\equiv \min_{W,H} \frac{1}{2}\|\mathcal{P}_\Omega\left(Y - WH^T\right)\|_F^2 + \frac{1}{2}\left\{\text{tr}(W^T L_w W) + \text{tr}(H^T L_h H)\right\} \quad (5)$$

where $L_w := \lambda_L\,\textbf{Lap}(E^w) + \lambda_w I_m$, and $L_h$ is defined similarly. Note that we subsume the regularization parameters in the definition of $L_w, L_h$. Note that $\|W\|_F^2 = \text{tr}(W^T I_m W)$.

The regularizer in (5) encourages solutions that are smooth with respect to the corresponding graphs. However, the Laplacian matrix can be replaced by other (positive, semi-definite) matrices that encourage structure by different means. Indeed, a very general class of Laplacian based regularizers was considered in [20], where one can replace $L_w$ by a function:

$$\langle \boldsymbol{x}, \tau(\textbf{Lap}(E))\boldsymbol{x}\rangle \quad \text{where} \quad \tau(\textbf{Lap}(E)) \equiv \sum_{i=1}^{|V|} \tau(\lambda_i)\boldsymbol{q}_i\boldsymbol{q}_i^T,$$

where $\{(\lambda_i, \boldsymbol{q}_i)\}$ constitute the eigen-system of $\textbf{Lap}(E)$ and $\tau(\lambda_i)$ is a scalar function of the eigenvalues. Our case corresponds to $\tau(\cdot)$ being the identity function. We briefly summarize other schemes that fit neatly into (5), apart from the graph regularizer we consider:

**Covariance matrices for variables:** [27] proposed a kernelized probabilistic matrix factorization (KPMF), which is a generative model to incorporate covariance information of the variables into matrix factorization. They assumed that each row of $W^\star, H^\star$ is generated according to a multivariate Gaussian, and solving the corresponding MAP estimation procedure yields exactly (5), with $L_w = C_w^{-1}$ and $L_h = C_h^{-1}$, where $C_w, C_h$ are the associated covariance matrices.

**Feature matrices for variables:** Assume that there is a feature matrix $X \in \mathbb{R}^{m \times d}$ for objects associated rows. For such $X$, one can construct a graph (and hence a Laplacian) using various methods such as k-nearest neighbors, $\epsilon$-nearest neighbors etc. Moreover, one can assume that there exists a kernel $k(\boldsymbol{x}_i, \boldsymbol{x}_j)$ that encodes pairwise relations, and we can use the Kernel Gram matrix as a Laplacian.

We can thus see that problem (5) is a very general scheme, and can incorporate information available in many different forms. In the sequel, we assume the matrices $L_w, L_h$ are given. In the theoretical analysis in Section 5, for ease of exposition, we further assume that the minimum eigenvalues of $L_w, L_h$ are unity. A general (nonzero) minimum eigenvalue will merely introduce multiplicative constants in our bounds.

## 3    GRALS: Graph Regularized Alternating Least Squares

In this section, we propose efficient algorithms for (5), which is convex with respect to $W$ or $H$ separately. This allows us to employ alternating minimization methods [25] to solve the problem. When $Y$ is fully observed, Li and Yeung [13] propose an alternating minimization scheme using block steepest descent. We deal with the partially observed setting, and propose to apply conjugate gradient (CG), which is known to converge faster than steepest descent, to solve each subproblem. We propose a very efficient Hessian-vector multiplication routine that results in the algorithm being highly scalable, compared to the (stochastic) gradient descent approaches in [14, 27].

We assume that $Y \in \mathbb{R}^{m \times n}$, $W \in \mathbb{R}^{m \times k}$ and $H \in \mathbb{R}^{n \times k}$. When optimizing $H$ with $W$ fixed, we obtain the following sub-problem.

$$\min_{H} f(H) = \frac{1}{2}\|\mathcal{P}_\Omega\left(Y - WH^T\right)\|_F^2 + \frac{1}{2}\text{tr}(H^T L_h H). \quad (6)$$

Optimizing $W$ while $H$ fixed is similar, and thus we only show the details for solving (6). Since $L_h$ is nonsingular, (6) is strongly convex.[2] We first present our algorithm for the fully observed case, since it sets the groundwork for the partially observed setting.

| **Algorithm 1** Hv-Multiplication for $g(\boldsymbol{s})$ | **Algorithm 2** Hv-Multiplication for $g_\Omega(\boldsymbol{s})$ |
|---|---|
| • **Given:** Matrices $L_h, W$<br>• **Initialization:** $\boldsymbol{G} = W^T W$<br>• **Multiplication:** $\nabla^2 g(\boldsymbol{s}_0)\boldsymbol{s}$:<br>  1 **Input:** $S \in \mathbb{R}^{n \times k}$ s.t.<br>    $\boldsymbol{s} = \text{vec}(S)$<br>  2 $A \leftarrow S\boldsymbol{G} + L_h S$<br>  3 **Return:** $\text{vec}(A)$ | • **Given:** Matrices $L_h, W, \Omega$<br>• **Multiplication:** $\nabla^2 g(\boldsymbol{s}_0)\boldsymbol{s}$:<br>  1 **Input:** $S \in \mathbb{R}^{k \times n}$ s.t.<br>    $\boldsymbol{s} = \text{vec}(S)$<br>  2 Compute $K = [\boldsymbol{k}_1, \ldots, \boldsymbol{k}_n]$ s.t.<br>    $\boldsymbol{k}_j \leftarrow \sum_{i \in \Omega_j} (\boldsymbol{w}_i^T \boldsymbol{s}_j) \boldsymbol{w}_i$<br>  3 $A \leftarrow K + S L_h$<br>  4 **Return:** $\text{vec}(A)$ |

## 3.1 Fully Observed Case

As in [5, 13] among others, there may be scenarios where $Y$ is completely observed, and the goal is to find the row/column embeddings that conform to the corresponding graphs. In this case, the loss term in (6) is simply $\|Y - WH^T\|_F^2$. Thus, setting $\nabla f(H) = 0$ is equivalent to solving the following Sylvester equation for an $n \times k$ matrix $H$:

$$HW^T W + L_h H = Y^T W. \tag{7}$$

(7) admits a closed form solution. However the standard Bartels-Stewart algorithm for the Sylvester equation requires transforming both $W^T W$ and $L_h$ into Schur form (diagonal in our case where $W^T W$ and $L_h$ are symmetric) by the QR algorithm, which is time consuming for a large $L_h$. Thus, we consider applying conjugate gradient (CG) to minimize $f(H)$ directly. We define the following quadratic function:

$$g(\boldsymbol{s}) := \frac{1}{2}\boldsymbol{s}^T M \boldsymbol{s} - \text{vec}(Y^T W)^T \boldsymbol{s}, \quad \boldsymbol{s} \in \mathbb{R}^{nk}, \quad M = I_k \otimes L_h + (W^T W) \otimes I_n$$

It is not hard to show that $f(H) = g(\text{vec}(H))$ and so we apply CG to minimize $g(\boldsymbol{s})$.

The most crucial step in CG is the Hessian-vector multiplication. Using the identity $(B^T \otimes A)\text{vec}(X) = \text{vec}(AXB)$, it follows that

$$(I_k \otimes L_h)\,\boldsymbol{s} = \text{vec}(L_h S), \quad \text{and} \quad ((W^T W) \otimes I_n)\,\boldsymbol{s} = \text{vec}(SW^T W),$$

where $\text{vec}(S) = \boldsymbol{s}$. Thus the Hessian-vector multiplication can be implemented by a series of matrix multiplications as follows.

$$M\boldsymbol{s} = \text{vec}(L_h S + S(W^T W)),$$

where $W^T W$ can be pre-computed and stored in $O(k^2)$ space. The details are presented in Algorithm 1. The time complexity for a single CG iteration is $O(nnz(L_h)k + nk^2)$, where $nnz(\cdot)$ is the number of non zeros. Since in most practical applications $k$ is generally small, the complexity is essentially $O(nnz(L_h)k)$ as long as $nk \leq nnz(L_h)$.

## 3.2 Partially Observed Case

In this case, the loss term of (6) becomes $\sum_{(i,j) \in \Omega}(Y_{ij} - \boldsymbol{w}_i^T \boldsymbol{h}_j)^2$, where $\boldsymbol{w}_i^T$ is the $i$-th row of $W$ and $\boldsymbol{h}_j$ is the $j$-th column of $H^T$. Similar to the fully observed case, we can define:

$$g_\Omega(\boldsymbol{s}) := \frac{1}{2}\boldsymbol{s}^T M_\Omega \boldsymbol{s} - \text{vec}(W^T Y)^T \boldsymbol{s},$$

where $M_\Omega = \bar{B} + L_h \otimes I_k$, $\bar{B} \in \mathbb{R}^{nk \times nk}$ is a block diagonal matrix with $n$ diagonal blocks $B_j \in \mathbb{R}^{k \times k}$. $B_j = \sum_{i \in \Omega_j} \boldsymbol{w}_i \boldsymbol{w}_i^T$, where $\Omega_j = \{i : (i,j) \in \Omega\}$. Again, we can see $f(H) = g_\Omega(\text{vec}(H^T))$. Note that the transpose $H^T$ is used here instead of $H$, which is used in the fully observed case.

For a given $\boldsymbol{s}$, let $S = [\boldsymbol{s}_1, \ldots \boldsymbol{s}_j, \ldots \boldsymbol{s}_n]$ be a matrix such that $\text{vec}(S) = \boldsymbol{s}$ and $K = [\boldsymbol{k}_1, \ldots, \boldsymbol{k}_j, \ldots, \boldsymbol{k}_n]$ with $\boldsymbol{k}_j = B_j \boldsymbol{s}_j$. Then $\bar{B}\boldsymbol{s} = \text{vec}(K)$. Note that since $n$ can be very large in practice, it may not be feasible to compute and store all $B_j$ in the beginning. Alternatively, $B_j \boldsymbol{s}_j$ can be computed in $O(|\Omega_j|k)$ time as follows.

$$B_j \boldsymbol{s}_j = \sum_{i \in \Omega_j} (\boldsymbol{w}_i^T \boldsymbol{s}_j) \boldsymbol{w}_i.$$

Thus $\bar{B}\boldsymbol{s}$ can be computed in $O(|\Omega|k)$ time, and the Hessian-vector multiplication $M_\Omega \boldsymbol{s}$ can be done in $O\left(|\Omega|k + nnz(L_h)k\right)$ time. See Algorithm 2 for a detailed procedure. As a result, each CG iteration for minimizing $g_\Omega(\boldsymbol{s})$ is also very cheap.

**Remark on Convergence.** In [2], it is shown that any local minimizer of (5) is a global minimizer of (5) if $k$ is larger than the true rank of the underlying matrix.[3] From [25], the alternating minimization procedure is guaranteed to globally converge to a block coordinate-wise minimum[4] of (5). The converged point might not be a local minimizer, but it still yields good performance in practice. Most importantly, since the updates are cheap to perform, our algorithm scales well to large datasets.

# 4 Convex Connection via Generalized Weighted Nuclear Norm

We now show that the regularizer in (5) can be cast as a generalized version of the weighted nuclear norm. The weights in our case will correspond to the scaling factors introduced on the matrices $W, H$ due to the eigenvalues of the shifted graph Laplacians $L_w, L_h$ respectively.

## 4.1 A weighted atomic norm:

From [7], we know that the nuclear norm is the gauge function induced by the atomic set: $\mathscr{A}_* = \{\boldsymbol{w}_i \boldsymbol{h}_i^T \; : \; \|\boldsymbol{w}_i\| = \|\boldsymbol{h}_i\| = 1\}$. Note that all rank one matrices in $\mathscr{A}_*$ have unit Frobenius norm. Now, assume $P = [\mathbf{p}_1, \ldots, \mathbf{p}_m] \in \mathbb{R}^{m \times m}$ is a basis of $\mathbb{R}^m$ and $S_p^{-1/2}$ is a diagonal matrix with $(S_p^{-1/2})_{ii} \geq 0$ encoding the "preference" over the space spanned by $\mathbf{p}_i$. The more the preference, the larger the value. Similarly, consider the basis $Q$ and the preference $S_q^{-1/2}$ for $\mathbb{R}^n$. Let $A = PS_p^{-1/2}$ and $B = QS_q^{-1/2}$, and consider the following "preferential" atomic set:

$$\mathscr{A} := \{\mathbb{O}_i = \boldsymbol{w}_i \boldsymbol{h}_i^T : \boldsymbol{w}_i = A\boldsymbol{u}_i, \boldsymbol{h}_i = B\boldsymbol{v}_i, \|\boldsymbol{u}_i\| = \|\boldsymbol{v}_i\| = 1\}. \tag{8}$$

Clearly, each atom $\mathbb{O}$ in $\mathscr{A}$ has non-unit Frobenius norm. This atomic set allows for biasing of the solutions towards certain atoms. We then define a corresponding atomic norm:

$$\|Z\|_{\mathscr{A}} = \inf \sum_{\mathbb{O}_i \in \mathscr{A}} |c_i| \quad \text{s.t. } Z = \sum_{\mathbb{O}_i \in \mathscr{A}} c_i \mathbb{O}_i. \tag{9}$$

It is not hard to verify that $\|Z\|_{\mathscr{A}}$ is a norm and $\{Z : \|Z\|_{\mathscr{A}} \leq \tau\}$ is closed and convex.

## 4.2 Equivalence to Graph Regularization

The graph regularization (5) can be shown to be a special case of the atomic norm (9), as a consequence of the following result:

**Theorem 1.** *For any $A = PS_p^{-1/2}$, $B = QS_q^{-1/2}$, and corresponding weighted atomic set $\mathscr{A}$,*

$$\|Z\|_{\mathscr{A}} = \inf_{W,H} \quad \frac{1}{2}\{\|A^{-1}W\|_F^2 + \|B^{-1}H\|_F^2\} \quad s.t. \quad Z = WH^T.$$

We prove this result in Appendix A. Theorem 1 immediately leads us to the following equivalence result:

**Corollary 1.** *Let $L_w = U_w S_w U_w^T$ and $L_h = U_h S_h U_h^T$ be the eigen decomposition for $L_w$ and $L_h$. We have*
$$\mathrm{Tr}\left(W^T L_w W\right) = \|A^{-1}W\|_F^2, \qquad and \quad \mathrm{Tr}\left(H^T L_h H\right) = \|B^{-1}H\|_F^2,$$

*where $A = U_w S_w^{-1/2}$ and $B = U_h S_h^{-1/2}$. As a result, $\|M\|_{\mathscr{A}}$ with the preference pair $(U_w, S_w^{-1/2})$ for the column space and the preference pair $(U_h, S_h^{-1/2})$ for row space is a weighted atomic norm equivalent for the graph regularization using $L_w$ and $L_h$.*

The results above allow us to obtain the dual weighted atomic norm for a matrix $Z$

$$\|Z\|_{\mathscr{A}}^* = \|A^T Z B\| = \|S_w^{-\frac{1}{2}} U_w^T Z U_h S_h^{-\frac{1}{2}}\| \tag{10}$$

which is a weighted spectral norm. An elementary proof of this result can be found in Appendix B. Note that we can then write

$$\|Z\|_{\mathscr{A}} = \|A^{-1}ZB^{-T}\|_* = \|S_w^{\frac{1}{2}}U_w^{-1}ZU_h^{-T}S_h^{\frac{1}{2}}\|_* \tag{11}$$

In [21], the authors consider a norm similar to (11), but with $A, B$ being diagonal matrices. In the spirit of their nomenclature, we refer to the norm in (11) as the *generalized* weighted nuclear norm.

## 5   Statistical Consistency in the Presence of Noisy Measurements

In this section, we derive theoretical guarantees for the graph regularized low rank matrix estimators. We first introduce some additional notation. We assume that there is a $m \times n$ matrix $Z^\star$ of rank $k$ with $\|Z^\star\|_F = 1$, and $N = |\Omega|$ entries of $Z^\star$ are uniformly sampled[5] and revealed to us (i.e., $Y = \mathcal{P}_\Omega(Z^\star)$). We further assume an one-to-one mapping between the set of observed indices $\Omega$ and $\{1, 2, \ldots, N\}$ so that the $t^{th}$ measurement is given by

$$y_t = Y_{i(t),j(t)} = \langle e_{i(t)}e_{j(t)}^T, Z^\star \rangle + \frac{\sigma}{\sqrt{mn}}\eta_t \qquad \eta_t \sim \mathcal{N}(0,1). \tag{12}$$

where $\langle \cdot, \cdot \rangle$ denotes the matrix trace inner product, and $i(t), j(t)$ is a randomly selected coordinate pair from $[m] \times [n]$. Let $A, B$ are corresponding matrices defined in Corollary 1 for the given $L_w, L_h$. W.L.O.G, we assume that the minimum singular value of both $L_w$ and $L_h$ is 1. We then define the following graph based complexity measures:

$$\alpha_g(Z) := \sqrt{mn}\frac{\|A^{-1}ZB^{-T}\|_\infty}{\|A^{-1}ZB^{-T}\|_F}, \qquad \beta_g(Z) := \frac{\|A^{-1}ZB^{-T}\|_*}{\|A^{-1}ZB^{-T}\|_F} \tag{13}$$

where $\|\cdot\|_\infty$ is the element-wise $\ell_\infty$ norm. Finally, we assume that the true matrix $Z^\star$ can be expressed as a linear combination of atoms from (8) (we define $\alpha^\star := \alpha_g(Z^\star)$):

$$Z^\star = AU^\star(V^\star)^T B^T, \;\; U^\star \in \mathbb{R}^{m \times k}, V^\star \in \mathbb{R}^{n \times k}, \tag{14}$$

Our goal in this section will be to characterize the solution to the following *convex* program, where the constraint set precludes selection of overly complex matrices in the sense of (13):

$$\hat{Z} = \arg\min_{Z \in \mathcal{C}} \frac{1}{N}\|\mathcal{P}_\Omega(Y-Z)\|_F^2 + \lambda\|Z\|_{\mathscr{A}} \text{ where } \mathcal{C} := \left\{ Z : \alpha_g(Z)\beta_g(Z) \leq \bar{c}_0\sqrt{\frac{N}{\log(m+n)}} \right\}, \tag{15}$$

where $\bar{c}_0$ is a constant depending on $\alpha^\star$.

A quick note on solving (15): since $\|\cdot\|_{\mathscr{A}}$ is a weighted nuclear norm, one can resort to proximal point methods [6], or greedy methods developed specifically for atomic norm constrained minimization [18, 22]. The latter are particularly attractive, since the greedy step reduces to computing the maximum singular vectors which can be efficiently computed using power methods. However, such methods will first involve computing the eigen decompositions of the graph Laplacians, and then storing the large, dense matrices $A, B$. We refrain from resorting to such methods in Section 6, and instead use the efficient framework derived in Section 3. We now state our main theoretical result:

**Theorem 2.** *Suppose we observe $N$ entries of the form* (12) *from a matrix $Z^\star \in \mathbb{R}^{m \times n}$, with $\alpha^\star := \alpha_g(Z^\star)$ and which can be represented using at most $k$ atoms from* (8). *Let $\hat{Z}$ be the minimizer of the convex problem* (15) *with $\lambda \geq C_1\sqrt{\frac{(m+n)\log(m+n)}{N}}$. Then, with high probability, we have*

$$\|\hat{Z} - Z^\star\|_F^2 \leq C\alpha^{\star 2}\max\{1, \sigma^2\}\frac{k(m+n)\log(m+n)}{N} + O\left(\frac{\alpha^{\star 2}}{N}\right)$$

*where $C, C_1$ are positive constants.*

See Appendix C for the detailed proof. A proof sketch is as follows:

**Proof Sketch:** There are three major portions of the proof:

- Using the fact that $Z^\star$ has unit Frobenius norm and can be expressed as a combination of at most $k$ atoms, we can show $\|Z^\star\|_\mathscr{A} \leq \sqrt{k}$ (Appendix C.1)
- Using (10), we can derive a bound for the dual norm of the gradient of the loss $\mathcal{L}(Z)$, given by $\|\nabla\mathcal{L}(Z)\|_\mathscr{A}^* = \|S_w^{-\frac{1}{2}}U_w^T \nabla\mathcal{L}(Z) U_h S_h^{-\frac{1}{2}}\|$. (Appendix C.2)
- Finally, using (13), we define a notion of restricted strong convexity (RSC) that the "error" matrices $Z^\star - \hat{Z}$ lie in. The proof follows closely along the lines of the equivalent result in [16], with appropriate modifications to accommodate our generalized weighted nuclear norm. (Appendix C.3).

### 5.1 Comparison to Standard Matrix Completion:

It is instructive to consider our result in the context of noisy matrix completion with uniform samples. In this case, one would replace $L_w, L_h$ by identity matrices, effectively ignoring graph information available. Specifically, the "standard" notion of spikiness ($\alpha_n := \sqrt{mn}\frac{\|Z\|_\infty}{\|Z\|_F}$) defined in [16] will apply, and the corresponding error bound (Theorem 2) will have $\alpha^\star$ replaced by $\alpha_n(Z^\star)$. In general, it is hard to quantify the relationship between $\alpha_g$ and $\alpha_n$, and a detailed comparison is an interesting topic for future work. However, we show below using simulations for various scenarios that the former is much smaller than the latter. We generate $m \times m$ matrices of rank $k = 10$, $M = U\Sigma V^T$ with $U, V$ being random orthonormal matrices and $\Sigma$ having diagonal elements picked from a uniform$[0,1]$ distribution. We generate graphs at random using the schemes discussed below, and set $Z = AMB^T$, with $A, B$ as defined in Corollary 1. We then compute $\alpha_n, \alpha_g$ for various $m$.

**Comparing $\alpha_g$ to $\alpha_n$:** Most real world graphs exhibit a power law degree distribution. We generated graphs with the $i^{th}$ node having degree $(m \times i^p)$ with varying negative $p$ values. Figure 1(a) shows that as $p \to 0$ from below, the gains received from using our norm is clear compared to the standard nuclear norm. We also observe that in general the weighted formulation is never worse then unweighted (The dotted magenta line is $\alpha_n/\alpha_g = 1$). The same applies for random graphs, where there is an edge between each $(i, j)$ with varying probability $p$ (Figure 1(b)).

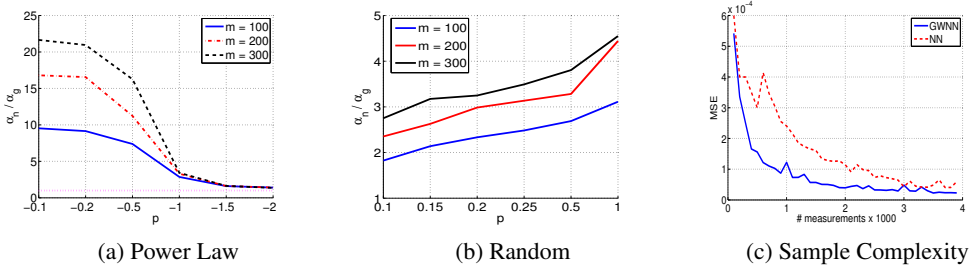

(a) Power Law    (b) Random    (c) Sample Complexity

Figure 1: (a), (b): Ratio of spikiness measures for traditional matrix completion and our formulation. (c): Sample complexity for the nuclear norm (NN) and generalized weighted nuclear norm (GWNN)

**Sample Complexity:** We tested the sample complexity needed to recover a $m = n = 200$, $k = 20$ matrix, generated from a power law distributed graph with $p = -0.5$. Figure 1(c) again outlines that the atomic formulation requires fewer examples to get an accurate recovery. We average the results over 10 independent runs, and we used [18] to solve the atomic norm constrained problem.

## 6 Experiments on Real Datasets

**Comparison to Related Formulations:** We compare GRALS to other methods that incorporate side information for matrix completion: the ADMM method of [12] that regularizes the entire target matrix; using known features (IMC) [10, 24]; and standard matrix completion (MC). We use the MOVIELENS $100k$ dataset,[6] that has user/movie features along with the ratings matrix. The dataset contains user features (such as age (numeric), gender (binary), and occupation), which we map

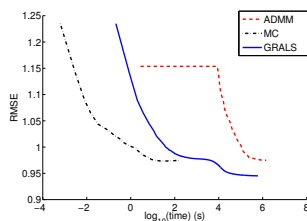

| Method | RMSE |
|---|---|
| IMC | 1.653 |
| Global mean | 1.154 |
| User mean | 1.063 |
| Movie mean | 1.033 |
| ADMM | 0.996 |
| MC | 0.973 |
| **GRALS** | **0.945** |

Figure 2: Time comparison of different methods on MOVIELENS 100k

Table 1: RMSE on the MOVIELENS dataset

Table 2: Data statistics.

| Dataset | # users | # items | # ratings | # links | rank used |
|---|---|---|---|---|---|
| Flixster ([11]) | 147,612 | 48,794 | 8,196,077 | 2,538,746 | 10 |
| Douban ([14]) | 129,490 | 58,541 | 16,830,839 | 1,711,802 | 10 |
| YahooMusic ([8]) | 249,012 | 296,111 | 55,749,965 | 57,248,136 | 20 |

into a 22 dimensional feature vector per user. We then construct a 10-nearest neighbor graph using the euclidean distance metric. We do the same for the movies, except in this case we have an 18 dimensional feature vector per movie. For IMC, we use the feature vectors directly. We trained a model of rank 10, and chose optimal parameters by cross validation. Table 1 shows the RMSE obtained for the methods considered. Figure 2 shows that the ADMM method, while obtaining a reasonable RMSE does not scale well, since one has to compute an SVD at each iteration.

**Scalability of GRALS:** We now demonstrate that the proposed GRALS method is more efficient than other state-of-the-art methods for solving the graph-regularized matrix factorization problem (5). We compare GRALS to the SGD method in [27], and GD: ALS with simple gradient descent. We consider three large-scale real-world collaborate filtering datasets with graph information: see Table 2 for details.[7] We randomly select 90% of ratings as the training set and use the remaining 10% as the test set. All the experiments are performed on an Intel machine with Xeon CPU E5-2680 v2 Ivy Bridge and enough RAM. Figure 3 shows orders of magnitude improvement in time compared to SGD. More experimental results are provided in the supplementary material.

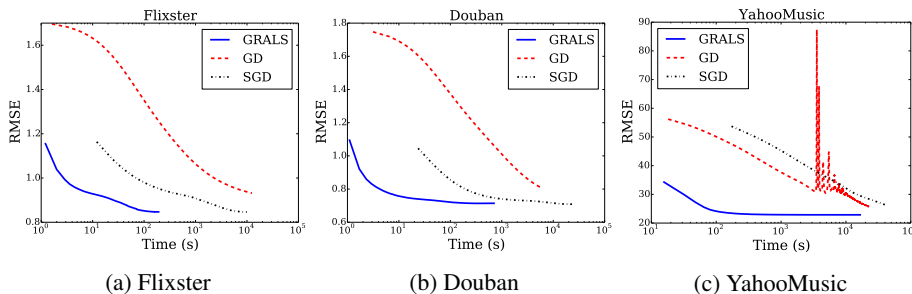

(a) Flixster     (b) Douban     (c) YahooMusic

Figure 3: Comparison of GRALS, GD, and SGD. The x-axis is the computation time in log-scale.

# 7 Discussion

In this paper, we have considered the problem of collaborative filtering with graph information for users and/or items, and showed that it can be cast as a generalized weighted nuclear norm problem. We derived statistical consistency guarantees for our method, and developed a highly scalable alternating minimization method. Experiments on large real world datasets show that our method achieves $\sim 2$ orders of magnitude speedups over competing approaches.

# Acknowledgments

This research was supported by NSF grant CCF-1320746. H.-F. Yu acknowledges support from an Intel PhD fellowship. NR was supported by an ICES fellowship.

## Footnotes

[1] The authors call this the "trust" between links in a social network

[2] In fact, a nonsingular $L_h$ can be handled using proximal updates, and our algorithm will still apply

[3]The authors actually show this for a more general class of regularizers.

[4]Nash equilibrium is used in [25].

[5]Our results can be generalized to non uniform sampling schemes as well.

[6]http://grouplens.org/datasets/movielens/

[7]See more details in Appendix D.

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
