[Supplementary Material · grmf-nips-supp.pdf]

## A    Proof of Theorem 1

**Proof**  For all $Z = \sum_i c_i \mathtt{a}_i$ with $\|A\|_{\mathscr{A}} = \sum_i |c_i|$, where $\mathtt{a}_i = A\boldsymbol{u}_i\boldsymbol{v}_i^T B^T$, we can construct the $i$-th column of $W$ and $H$ as

$$\boldsymbol{w}_i = \sqrt{|c_i|}A\boldsymbol{u}_i \quad \text{and} \quad \boldsymbol{h}_i = \sqrt{|c_i|}B\boldsymbol{v}_i.$$

Clearly, we have $Z = WH^T$ and

$$\|A^{-1}W\|_F^2 = \|B^{-1}H\|_F^2 = \sum_i |c_i|$$

Thus, if follows that LHS $\geq$ RHS. Oh the other hand, for a matrix $Z = WH^T$, we can construct

$$\boldsymbol{u}_i = \frac{A^{-1}\boldsymbol{w}_i}{\|A^{-1}\boldsymbol{w}_i\|} \quad \text{and} \quad \boldsymbol{v}_i = \frac{B^{-1}\boldsymbol{h}_i}{\|B^{-1}\boldsymbol{h}_i\|},$$

and $c_i = \|A^{-1}\boldsymbol{w}_i\|\|B^{-1}\boldsymbol{h}_i\|$. Clearly, we have $\boldsymbol{w}_i\boldsymbol{h}_i^T = c_i A\boldsymbol{u}_i\boldsymbol{v}_i B^T$ and $Z = \sum_i c_i A\boldsymbol{u}_i\boldsymbol{v}_i^T B^T$. We also have

$$|c_i| = \|A^{-1}\boldsymbol{w}_i\|\|B^{-1}\boldsymbol{h}_i\| \leq \frac{1}{2}\left(\|A^{-1}\boldsymbol{w}_i\|^2 + \|B^{-1}\boldsymbol{h}_i\|^2\right)$$

by AM-GM inequality. Thus, we have LHS $\leq$ RHS. $\blacksquare$

## B    Dual Weighted Generalized Nuclear Norm

Recall the definition of the weighted atomic set:

$$\mathscr{A} := \{\mathtt{a}_i = \boldsymbol{w}_i\boldsymbol{h}_i^T : \boldsymbol{w}_i = A\boldsymbol{u}_i, \boldsymbol{h}_i = B\boldsymbol{v}_i, \|\boldsymbol{u}_i\| = \|\boldsymbol{v}_i\| = 1\}.$$

We derive the dual norm as follows.

$$\begin{aligned}
\|Z\|_{\mathscr{A}}^* &= \sup_{\mathtt{a}\in\mathscr{A}} \langle \mathtt{a}, Z\rangle \\
&= \sup_{\boldsymbol{u},\boldsymbol{v}} \langle A\boldsymbol{u}\boldsymbol{v}^T B^T, Z\rangle, \quad \text{s.t. } \|\boldsymbol{u}\| = \|\boldsymbol{v}\| = 1 \\
&= \sup_{\boldsymbol{u},\boldsymbol{v}} \text{Tr}\left(B\boldsymbol{v}\boldsymbol{u}^T A^T Z\right), \quad \text{s.t. } \|\boldsymbol{u}\| = \|\boldsymbol{v}\| = 1 \\
&= \sup_{\boldsymbol{u},\boldsymbol{v}} \boldsymbol{u}^T A^T Z B\boldsymbol{v}, \quad \text{s.t. } \|\boldsymbol{u}\| = \|\boldsymbol{v}\| = 1 \\
&= \|A^T Z B\|
\end{aligned}$$

## C    Proof of Theorem 2

The proof of our main Theorem 2 follows the similar steps used in [16]. The main idea is to use Theorem 3 [17] to obtain the consistency guarantee. Our proof steps (and indeed that of [16]) are a consequence of carefully bounding the various quantities needed to make Theorem 3 hold:

**Theorem 3** (Theorem 1 of [17]). *For the convex optimization problem of the following form:*

$$\hat{Z} = \arg\min_{Z\in\mathbb{R}^{m\times n}} \mathcal{L}(Z; X_1, \ldots, X_N) + \lambda\mathcal{R}(Z),$$

*where*

- (a) *the regularizer $\mathcal{R}$ is a norm and is **decomposable** with respect to the subspace pair $(\mathcal{M}, \bar{\mathcal{M}}^\perp)$, where $\mathcal{M} \subseteq \bar{\mathcal{M}}$ is a subspace.*

- (b) *the loss function $\mathcal{L}$ is convex and differentiable, and satisfies **restricted strong convexity** with curvature $\kappa$ and tolerance $\tau$*

*with a strictly positive regularization constant $\lambda \geq 2\mathcal{R}^* (\nabla\mathcal{L}(Z^\star))$, any optimal solution $\hat{Z}$ satisfies the bound*

$$\|\hat{Z} - Z^\star\|^2 \leq 9\frac{\lambda^2}{\kappa^2}\Psi(\mathcal{M})^2 + \frac{\lambda}{\kappa}\left\{2\tau^2(Z^\star) + 4\mathcal{R}(Z^\star_{M^\perp})\right\}, \tag{C.1}$$

*where $\Psi(\mathcal{M}) := \sup_{Z\in\mathcal{M}\backslash\{0\}}\frac{\mathcal{R}(Z)}{\|Z\|_F}$. Furthermore, if $Z^\star \in \mathcal{M}$, then the bound becomes*

$$\|\hat{Z} - Z^\star\|^2 \leq 9\frac{\lambda^2}{\kappa^2}\Psi(\mathcal{M})^2. \tag{C.2}$$

See [17] for the detailed definitions of **decomposable norms** and **restricted strong convexity**.

To apply Theorem 3 to analyze the consistency of (15), we make the following remarks:

- $\mathcal{R}(Z) = \|Z\|_\mathscr{A}$: the weighted atomic norm defined in (9).
- $\mathcal{R}^*(Z) = \|Z\|_\mathscr{A}^*$: the dual norm of the weighted atomic norm.
- $\mathcal{M} = \{Z = AMB^T : \text{rank}(Z) = k\}$: the subspace we are interested in.
- $\mathcal{L}(Z; X_1, \ldots, Z_n) = \frac{1}{N}\sum_{i=1}^N (y_t - \langle X_t, Z\rangle)^2$: where $X_t = e_{i(t)}e_{j(t)}^T$ (See the corresponding measurement model in (12)). Because the squared-$L_2$ loss is used in our setting, the restricted strong convexity parameter $\kappa$ is related to the minimum singular value of the Hessian of $\mathcal{L}(Z; X_1, \ldots, X_N)$. Thus, from (C.1) and (C.2) we can see that the bounds remain the same when we scale $\{X_t\}$ and $\{y_t\}$ by the same constant as both $\kappa$ and the lower bound of $\lambda$ (which is $\mathcal{R}^*(\nabla\mathcal{L}(Z^\star))$) are scaled with the same constant. Thus, in the following proof, we consider the following equivalent statistical measurement model:

$$y_t = \langle\sqrt{mn}\epsilon_t e_{i(t)}e_{j(t)}^T, Z^\star\rangle + \sigma\epsilon_t\eta_t \tag{C.3}$$

where $\epsilon_t$ are i.i.d. Rademacher random variables [16]. Let's re-define

$$X_t := \sqrt{mn}\epsilon_t e_{i(t)}e_{j(t)}^T, \tag{C.4}$$
$$y_t := \langle X_t, Z\rangle + \sigma\epsilon_t\eta_t.$$

In addition, we also define $\mathscr{X}(Z) \in \mathbb{R}^N$ be the vector such that $\mathscr{X}(Z)_t = \langle X_t, Z\rangle$.

- The exact restricted strong convexity condition we need for (15) is as follows:

$$\frac{1}{\sqrt{N}}\|\mathscr{X}(Z)\| \geq \frac{1}{8}\|Z\|_F\left\{1 - \hat{c}_0\frac{\alpha_g(Z)}{\sqrt{N}}\right\} \quad \forall Z \in \mathcal{C}, \tag{C.5}$$

where $\mathcal{C}$ is defined in (15) and $\hat{c}_0$ is a constant (similar to [16, Eq. 28]).

In the following subsections, we prove bounds for the quantities needed for establishing Theorem 2 via the following steps:

- In Section C.1, we derive an upper bound for $\Psi(\mathcal{M})$.
- In Section C.2, we derive an upper bound for $\mathcal{R}^*(\nabla\mathcal{L}(Z^*))$.
- In Section C.3, we prove that the restricted strong convexity (C.5) holds $\mathcal{L}$ with exponentially high probability.

Note that for the sake of proving our results, we assume that the target matrix $Z^\star$ is exactly low rank, and the minimum singular values of $A$, $B$ are 1. Our results can be extended in a straightforward manner when $Z^\star$ does not exactly lie in $\mathcal{M}$ (it is approximately low rank).

## C.1 Bounding the Atomic Norm

Based on the definition of $\Psi(\mathcal{M})$, we can derive its upper bound on the atomic norm of $Z \in \mathcal{M}$ with $\|Z\|_F = 1$.

**Lemma 1.** *Let $Z \in \mathbb{R}^{m\times n}$, $Z = AUV^TB^T$, $rank(Z) = k$ be a linear combination of atoms in $\mathscr{A}$. Then, with the assumption $\|Z\|_F = 1$ we have*

$$\|Z\|_\mathscr{A} \leq \sqrt{k}$$

**Proof**

$$\|Z\|_{\mathscr{A}} = \|UV^T\|_* \leq \sqrt{k}\|UV^T\|_F \leq \sqrt{k}\|A^{-1}AUV^TB^{-T}\|_F,$$

where the first inequality follows from Cauchy Schwartz, and the second inequality follows from noting that $\|A^{-1}\| \leq 1$ and likewise for $\|B^{-1}\|$, since we assumed that the minimum singular value of both $L_w, L_h$ is unity. ∎

## C.2 Bounding the Dual Norm of the Gradient of Loss Function

A key ingredient for our main result will be a bound on the dual norm of the gradient of the loss function, which we will use to bound the regularizer $\lambda$. From Eq. (11), and our problem set up in Eq. (16), we have the following set of inequalities:

$$\|\nabla\mathcal{L}\|_{\mathscr{A}}^* = \|S_w^{-\frac{1}{2}}U_w^T\nabla\mathcal{L}U_hS_h^{-\frac{1}{2}}\| \overset{(i)}{\leq} \|S_w^{-\frac{1}{2}}U_w^T\|\|U_hS_h^{-\frac{1}{2}}\|\|(\nabla\mathcal{L})\|$$

$$= \frac{\|\nabla\mathcal{L}\|}{\sigma_{min}(L_w^{\frac{1}{2}})\sigma_{min}(L_h^{\frac{1}{2}})} \overset{(ii)}{\leq} C\sigma\sqrt{\frac{(m+n)\log(m+n)}{N}}, \tag{C.6}$$

with probability at least $1 - \exp\left(c\sqrt{N\log(m+n)}\right)$. $(i)$ appeals to submultiplicativity, and we prove $(ii)$ below. From our assumption about unit minimum singular values, we can ignore the denominator.

Here we develop a bound on the spectral norm of the gradient of the loss function, specifically step $(ii)$ in (C.6). Our proof follows that of the corresponding result in [16], which we show here for completeness.

Recall the definition of $X_t := \sqrt{mn}\epsilon_t e_{i(t)}e_{j(t)}^T$ in (C.4), we have the gradient of the loss function given by

$$\nabla\mathcal{L} = \frac{\sigma}{N}\left\|\sum_{t=1}^N \eta_t X_t\right\| \tag{C.7}$$

For ease of exposition, assume $m = n$. We now show that with high probability, the quantity in (C.7) is bounded above by $C\sigma\sqrt{\frac{m\log(m)}{N}}$. For $m \neq n$, our bound can be made necessarily better since the result we prove can be seen as holding for $\max\{m, n\}$. To prove our result, we make use of the matrix noncommutative Bernstein inequality (Theorem 3.2 in [19]):

**Lemma 2.** *Let $X_1, \cdots, X_N$ be independent, zero mean random matrices of size $m \times n$. Suppose $\rho_t^2 := \max\{\|\mathbb{E}[X_tX_t^T]\|, \|\mathbb{E}[X_t^TX_t]\|\}$, and suppose $\|X_t\| \leq \bar{M}$ almost surely $\forall t$. Then for any $\tau > 0$*

$$\mathbb{P}\left[\left\|\sum_{t=1}^N X_t\right\| > \tau\right] \leq (m+n)\exp\left(\frac{-\frac{\tau^2}{2}}{\sum_{t=1}^N \rho_t^2 + \frac{\bar{M}\tau}{3}}\right)$$

The above result holds even for sub-exponential random variables [23] and the Orlicz norm instead of the spectral norm being bounded above by a constant $\bar{M}$.

To use Lemma 2, we first derive bounds on the relevant quantities. First, note that for all $t$, $X_t$ has a single non zero entry of magnitude $m$. Noting that $\eta_t$ is a standard Gaussian random variable, we can bound the Orlicz norm $\|\eta_t X_t\|_{\psi 1} \leq m$. Also

$$\mathbb{E}\left[\eta_t^2 X_t^T X_t\right] = \mathbb{E}\left[m^2 e_{j(t)}e_{i(t)}^T e_{i(t)}e_{j(t)}^T\right] = m^2\mathbb{E}\left[e_{j(t)}e_{j(t)}^T\right]$$

The matrix inside the expectation has a 1 in the $j(t), j(t)$ location. Since $j(t)$ is chosen uniformly at random, the expected value of the non zero entry is $1/m$. This means

$$\|\mathbb{E}\left[\eta_t^2 X_t^T X_t\right]\| = m = \|\mathbb{E}\left[\eta_t^2 X_t X_t^T\right]\|$$

This gives $\bar{M} = \rho_t^2 = m$. Setting $\tau = N\delta$, and from Lemma 2, we get

$$\mathbb{P}\left[\frac{\sigma}{N}\left\|\sum_{t=1}^N \eta_t X_t\right\| > \sigma\delta\right] \leq 2m\exp\left(-\frac{CN\delta}{m}\right)$$

Our result then follows by setting $\delta = c\sqrt{\frac{m\log(m)}{N}}$.

## C.3 Restricted Strong Convexity for Generalized Weighted Nuclear Norm

The proof of this result mirrors the corresponding proof in [16]. Hence, to keep things simple, we skip the steps that are common between our method and [16], and only pause to highlight the differences.

First, note that since we assume uniformly weighted samples, we need not concern ourselves with the "weight" matrices that are considered in [16]. Also, define $\mathscr{X}(Z)_t = \langle X_t, Z \rangle$, where $X_t$ is defined as in Appendix C.2. Then, the RSC condition requires us to prove (C.5), which we re-state it again as follows.

$$\frac{1}{\sqrt{N}}\|\mathscr{X}(Z)\| \geq \frac{1}{8}\|Z\|_F \left\{ 1 - \hat{c}_0 \frac{\alpha_g(Z)}{\sqrt{N}} \right\} \quad \forall Z \in \mathcal{C},$$

where $\mathcal{C}$ is defined in (15) and $\hat{c}_0$ is a constant In other words, we wish to prove that the following event holds with high probability:

$$\mathcal{E}_1 := \left\{ \forall Z \in \mathcal{C} : \frac{1}{\sqrt{N}}\|\mathscr{X}(Z)\| \geq \frac{1}{8}\|Z\|_F - \frac{\hat{c}_0 m}{8\sqrt{N}}\|Z\|_F \frac{\|M\|_\infty}{\|M\|_F} \right\}, \tag{C.8}$$

where $M := A^{-1}ZB^{-T}$ and $\alpha_g$ is the spikiness defined in (13). Subtracting $\|Z\|_F$ from both sides of the inequality in (C.8), we get

$$\frac{1}{\sqrt{N}}\|\mathscr{X}(Z)\| - \|Z\|_F \geq -\frac{7}{8}\|Z\|_F - \frac{\hat{c}_0 m}{8\sqrt{N}}\|Z\|_F \frac{\|M\|_\infty}{\|M\|_F},$$

and hence we can define a "bad" event as

$$\mathcal{E}_2 := \left\{ \exists Z \in \mathcal{C} : \left| \frac{1}{\sqrt{N}}\|\mathscr{X}(Z)\| - \|Z\|_F \right| > \frac{7}{8}\|Z\|_F + \frac{\hat{c}_0 m}{8\sqrt{N}}\|Z\|_F \frac{\|M\|_\infty}{\|M\|_F} \right\} \tag{C.9}$$

Now, due to the definition of $\mathcal{C}$, event $\mathcal{E}_2$ is invariant under rescaling of $Z$ (so as $M := A^{-1}ZB^{-T}$). Thus, without loss of generality, we may assume that $\|M\|_\infty = 1/m$. Then, the remaining degrees of freedom in the set $\mathcal{C}$ can be parameterized in terms of the quantities $D = \|M\|_F$ and $\rho = \|M\|_*$. For any $Z = AMB^T \in \mathcal{C}$ with $\|M\|_\infty = 1/d$ and $\|M\|_F \leq D$, we have $\|M\|_* \leq \rho(D)$, where

$$\rho(D) := \bar{c}_0 D^2 \left( \frac{N}{m\log(m)} \right)^{\frac{1}{2}}.$$

For each radius $D > 0$, consider the set

$$\mathscr{B}(D) := \left\{ Z = AMB^T : \|M\|_\infty = 1/m, \|M\|_F \leq D, \|M\|_* \leq \rho(D) \right\}, \tag{C.10}$$

and consider the event

$$\mathcal{E}_3(D) := \left\{ \exists Z \in \mathscr{B}(D) : \left| \frac{1}{\sqrt{N}}\|\mathscr{X}(Z)\| - \|Z\|_F \right| > \frac{3}{4}D + \frac{\hat{c}_0 m}{8\sqrt{N}}\|Z\|_F \frac{\|M\|_\infty}{\|M\|_F} \right\} \tag{C.11}$$

Now, note that the RHS of inequality in the above event satisfies, for $Z \in \mathscr{B}(D)$

$$\frac{3}{4}D + \frac{\hat{c}_0 m}{8\sqrt{N}}\|Z\|_F \frac{\|M\|_\infty}{\|M\|_F} = \frac{3}{4}D + \frac{\hat{c}_0}{8\sqrt{N}}\frac{\|Z\|_F}{\|M\|_F}$$

$$\geq \frac{3}{4}D + \frac{\hat{c}_0}{8\sqrt{N}},$$

where the first equality follows since $Z = AMB^T \in \mathcal{C} \Rightarrow \|M\|_\infty = 1/m$, and the last inequality follows since $\|Z\|_F = \|AMB^T\|_F \geq \sigma_{min}(A)\sigma_{min}(B)\|M\|_F$, and noting that the minimum singular values of $A, B$ are unity. Finally, we define the event

$$\mathcal{E}_4(D) := \left\{ \exists Z \in \mathscr{B}(D) : \left| \frac{1}{\sqrt{N}}\|\mathscr{X}(Z)\| - \|Z\|_F \right| \geq \frac{3}{4}D + \frac{\hat{c}_0}{8\sqrt{N}} \right\} \tag{C.12}$$

Let $\mathcal{S}_1$ be the set of $Z$ that satisfy event $\mathcal{E}_1$, and similarly define sets $\mathcal{S}_2, \mathcal{S}_3(D), \mathcal{S}_4(D)$. The following statement will be used to prove our results: for each fixed $D > 0$,

$$\mathcal{S}_4(D) \supset \mathcal{S}_3(D) \supset \mathcal{S}_2 \supset \mathcal{S}_1^c \tag{C.13}$$

Meaning that if we can show that $\mathcal{E}_4$ holds with very low probability for a fixed D, then it follows from (C.13) that $\mathcal{E}_1$ holds with high probability. The remainder of the proof will focus on doing so.

First, note that the event $\mathcal{E}_4$ defined in (C.12) is exactly the same as the event defined in [16, Eq. 29]. Hence, we can use the exact same argument as described in [16, Section 5.2] to obtain

$$\mathbb{P}(\mathcal{E}_4(D)) \leq c_1 \exp\left(-c_2 N D^2\right).$$

Now, we have the following result:

**Lemma 3.** *Suppose there are constants $c_1, c_2$ so that, for each fixed $D > 0$,*

$$\mathbb{P}(\mathcal{E}_4(D)) \leq c_1 \exp\left(-c_2 N D^2\right)$$

*then $\exists$ a universal constant $c_2'$ so that*

$$\mathbb{P}(\mathcal{E}_2) \leq c_1 \frac{\exp(-c_2' m \log(m))}{1 - \exp(-c_2' m \log(m))}.$$

The statement is the same as [16, Lemma 3], but we have to slightly modify the proof to adapt it to our setting. We do this in Appendix C.4.

Lemma 3 allows us to shows that if $\mathcal{E}_4$ holds with low probability, then $\mathcal{E}_2$ holds with low probability as well. Since by construction, $\mathcal{E}_1^c \subset \mathcal{E}_2$, the RSC result follows.

Since the results derived here are for the statistical model defined by (C.3), we go from this model to the initial model that we consider in (12). To this end, one needs to make the following two transformations, as explained in the remarks following Theorem 3:

- Scale the magnitude of $X_t$, and consequently $\lambda$ by $1/m$
- Scale the noise variance $\sigma$ by $m$.

The rates we obtain in Theorem 2 remain unchanged as a result of this scaling.

### C.4 Proof of Lemma 3

The proof is similar to [16, Lemma 3], we include it with our notation for completeness. For any $Z = A M B^T \in \mathcal{C}$, with $\|M\|_\infty = 1/m$, based on the definition of $\mathcal{C}$ in (15), we have

$$\|M\|_F^2 \geq \bar{c}_0^{-1} \|M\|_* \left(\frac{m \log(m)}{N}\right)^{\frac{1}{2}} \geq \bar{c}_0^{-1} \|M\|_F \left(\frac{m \log(m)}{N}\right)^{\frac{1}{2}},$$

which gives us $\|M\|_F \geq \bar{c}_0^{-1} \left(\frac{m \log(m)}{N}\right)^{\frac{1}{2}}$. Hence, we only need to focus on sets $\mathscr{B}(D)$ where $D > \mu := \bar{c}_0^{-1} \left(\frac{m \log(m)}{N}\right)^{\frac{1}{2}}$. For $l = 1, 2, \dots$ and $a = \frac{7}{6}$ define

$$S_l := \left\{Z = A M B^T \in \mathcal{C} : \|M\|_\infty = 1/m, \ a^{l-1} \mu \leq \|M\|_F \leq a^l \mu, \ \|M\|_* \leq \rho(a^l \mu)\right\}$$

From the definition of (C.9), we have $S_l \subset \mathscr{B}(a^l \mu)$. Now, if $\mathcal{E}_2$ holds for some $Z$, then $Z$ must belong to $S_l$ for some $l$. When $Z \in S_l$, we know $\exists Z \in \mathscr{B}(a^l \mu)$ such that

$$\begin{aligned}
\left|\frac{\|\mathscr{X}(Z)\|}{\sqrt{N}} - \|Z\|_F\right| &\geq \frac{7}{8}\|Z\|_F + \frac{\hat{c}_0 m}{8\sqrt{N}}\|Z\|_F \frac{\|M\|_\infty}{\|M\|_F} \\
&\geq \frac{7}{8}\|Z\|_F + \frac{\hat{c}_0}{8\sqrt{N}} \\
&\geq \frac{7}{8} a^{l-1} \mu + \frac{\hat{c}_0}{8\sqrt{N}} \\
&= \frac{3}{4} a^l \mu + \frac{\hat{c}_0}{8\sqrt{N}} \qquad \text{since a = 7/6.}
\end{aligned}$$

Thus, we have shown that when this $Z \in S_l$, then $\mathcal{E}_4(a^l\mu)$ must hold. Because any $Z$ which make the event $\mathcal{E}_2$ hold must fall into some set $S_l$, the union bound implies that

$$\mathbb{P}[\![\mathcal{E}_2]\!] \leq \sum_{l=1}^{\infty} \mathbb{P}[\![\mathcal{E}(a^l\mu)]\!]$$

$$\leq c_1 \sum_{l=1}^{\infty} \exp\left(-c_2 N a^{2l}\mu^2\right)$$

$$\leq c_1 \sum_{l=1}^{\infty} \exp\left(-2c_2 \log(a) l N \mu^2\right)$$

$$\leq c_1 \frac{\exp(-\bar{c}_2 N \mu^2)}{1 - \exp(-\bar{c}_2 N \mu^2)}$$

$$= c_1 \frac{\exp(-c_2' m \log m)}{1 - \exp(-c_2' m \log m)},$$

where the last equality follows as $N\mu^2 = \bar{c}_0^{-1}(m \log m)$.

## D  Additional Details for Experimental Results

**Experimental environment and Implementation.** All the experiments are generated on an Intel machine with 2 Xeon CPU E5-2680 v2 Ivy Bridge and 256 GB ram. GRALS is implemented using a MEX routine written in C++. For SGD and GD, we optimize the code from [27] in several ways: vectorization of for-loops and parallel residual computation using a MEX routine using C++. All the implementations employ embarrassing parallelization for BLAS operations whenever applicable (either through parallel BLAS library in Matlab, or simple OpenMP parallel-for loop).

**Parameters.** In Section 6, we show the results in Figure 3 to demonstrate the superiority of the proposed algorithm GRALS over the existing approaches: SGD and GD [27]. In Table Supp-1, we list the parameters used to generate the results. Note that in all the datasets we used, there is only one set of variables which comes with graph information (say $W$). Thus, the regularization consists of three terms as follows:

$$\lambda_L \operatorname{Tr}\left(W^T \mathbf{Lap}(E^w)W\right) + \lambda_w \|W\|_F^2 + \lambda_h \|H\|_F^2.$$

In addition to the regularization parameters, there are algorithmic parameters for each approach:

- GRALS: the number of CG iterations to solve each sub-problem
- SGD: the learning rate, $\eta_{sgd}$
- GD: the learning rate, $\eta_{gd}$

In Table Supp-1, we also report the best algorithm-specific parameters for each method.

Table Supp-1: Parameters used in the experiments for Figure 3

| | $\lambda_L$ | $\lambda_w$ | $\lambda_h$ | GRALS CG-iters | SGD $\eta_{sgd}$ | GD $\eta_{gd}$ |
|---|---|---|---|---|---|---|
| Flixster | 0.01 | 0.01 | 0.02 | 3 | $10^{-4}$ | $10^{-6}$ |
| Douban | 1 | 0.01 | 1.01 | 5 | $10^{-4}$ | $10^{-6}$ |
| YahooMusic | 100 | 100 | 200 | 20 | $10^{-6}$ | $10^{-6}$ |

**Graph Information in Datasets.** For Flixster and Douban, the datasets come with the graph information among users. For YahooMusic, we use the Yahoo Music Track 2 dataset from KDDCup 2011 [8] for the purpose of showing that GRALS scales much better than other approaches. As most of entries in the test split of the Track 2 dataset are marked as $-1$ (for the classification purpose in that track), we only use the training set in our experiments. The original training set is randomly partitioned into a $90 - 10$ training-test split. There is no explicit graph information in YahooMusic. Thus, we use the provided "album", "artist", and "genre" attributes for each item (or music track)

to construct a binary indicator vector and construct a 10-NN graph graph using the inner product distance over all the items.

**RMSE Performance.** Because the aim of this paper is to develop scalable algorithms and consistency results for graph regularized matrix factorization (4), we did not include the performance comparison table (similar to Table 1) for other large datasets for want of space. Here, we report the results in Table Supp-2. Note that there are other approaches to incorporate graph information into collaborative filtering, which might lead to different RMSEs. A detailed comparison to all such methods is beyond the scope of this paper. However, whenever there is a means to incorporate pairwise relationships between user-user variables or item-item variables, we can use GRALS to achieve the same results as other approaches, but at a much faster rate. Note that the Yahoo Music dataset has ratings in the range $[0, 100]$ and hence the larger RMSE values. A fairer comparison can be obtained by dividing the results by 20, to correspond to ratings in the range $[0, 5]$.

Table Supp-2: RMSE of various methods on the datasets considered in Figure 3. PMF : Our method with graph Laplacians replaced by identity matrices.

| DATASET | PMF | GRALS | Global Mean | User Mean | Item Mean |
|---------|-----|-------|-------------|-----------|-----------|
| Flixster | 0.923 | 0.845 | 1.092 | 0.979 | 1.088 |
| Douban | 0.719 | 0.714 | 0.907 | 0.848 | 0.790 |
| YahooMusic | 23.823 | 22.872 | 37.941 | 43.308 | 38.042 |