[Reviews · NeurIPS 2015]

Submitted by Assigned_Reviewer_1

The authors propose a method for incorporating underlying graph structure among users/items in a collaborative filtering setting. The novelty of this method is that it is the first method to both incorporate graph structure and deal with sparse data.

The method itself is fairly straightforward and clearly explained. It amounts to block-coordinate ascent, solving a convex problem at each iteration. One potential methodological contribution is to solve for the the low-rank factors using conjugate-gradient methods to avoid slow matrix decompositions, but it's unclear whether this is the first time such an approach was proposed.

Connections are drawn to the nuclear norm minimization of Srebro and Salakhutdinov (2010), and the authors show that their method is equivalent to employing a weighted atomic norm, where the weights are derived from the graph structure. This connection allows them to derive a consistency bound on recovering the true low-rank matrix. The bound is specified in terms of a quantity \alpha_^* which is shown empirically to be much smaller for realistic matrices than the corresponding bound for matrix completion.

Finally, the method is validated on three real datasets and is shown to outperform standard matrix completion in that it is able to achieve a given RMSE in much less time.

The work is of high quality and clearly explained. I amn less confident of its originality but if this is indeed the first time the conjugate-gradient method embedded in the coordinate-ascent scheme was emplyed in a collaborative filtering setting, then this is a reasonably novel contribution. Overall, this seems to be a signficiant advancement in how further exploiting known structure in features to improve collaborative filtering models.

Summary: This paper represents a substantial contribution to the growing body of work around leveraging underlying structure among users/items for collaborative filtering. The paper is well-written and the contributions are both theoretical (consistency bounds in terms of a quantity that is empirically verified to be tighter) and practical in nature (faster method by orders of magnitude, superior performance on real data sets).

Submitted by Assigned_Reviewer_2

Paper 1260 tackles the problem of matrix completion with side information represented as a graph. The authors claim that this is a general framework in that the side information can come from many different kinds of sources. Major contributions of this paper are scalable learning algorithm for matrix completition with graph structures, connecting it to existing works in matrix factorization literature, and verifying the proposed method theoretically as well as experimentally.

As the authors claim, this problem is widely applicable such as recommendation systems with social network information. Considering the size of such dataset, it is important to bulid a scalable algorithm as the authors proposed.

This paper is well-structured and well-written. It is good to have both theoretical analysis of consistency and experiments with real data. Also, it is useful to connect the proposed method in the context of wider convex optimization and noisy matrix approximation.

Regarding the experiment (Section 6): some baselines in Table 1 are too simple; no one might be interested in comparison with Global/User/Movie mean. I suggest comparing against more recent, state-of-the-art methods such as

Salakhutdinov and Mnih, Probabilistic Matrix Factorization [NIPS 2008], Mackey et al., Divide-and-Conquer Matrix Factorization [NIPS 2011], Lee et al., Local Low-Rank Matrix Approximation [ICML 2013].

It is impressive that the proposed method runs in O(10^3) seconds with dataset in Table 2. However, to claim scalability, I recommend running the algorithm on larger dataset, such as Netflix or Yahoo Music. This way it is also possible comparing RMSE score with other methods as well.

Minor comments: 1) Why Figure 1 and Figure 2 have different format of x-axis? I recommend using format of Figure 2 (10^1, 10^2, ...) instead of plotting log(time) directly. Log-scale without specifying base is ambiguous. 2) Citations are not in NIPS format.
Summary: This paper presents GRALS, an efficient way of optimizing graph-regularized matrix completion problem. With some minor issues with experiments, I see this paper is generally well-written and clear.

Submitted by Assigned_Reviewer_3

The paper considers matrix completion in a setting where the row and column variables lie on graphs. The authors

develop a scalable alternating least squares algorithm. Further, they show that the regularizer in the optimization problem can be seen as a generalized form of weighted nuclear norm, and derive statistical consistency guarantees for the low rank matrix estimators. Experiments comparing to leading methods on a movie ratings data set shows their method achieving lowest RMSE. Further, their ALS algorithm is shown to scale orders of magnitude better than SGD.

a. The description of how the row/column graphs are generated from the movielens dataset is vague in the paper; please clarify.

b. Sections 5 and 5.1 were difficult to follow. Terms such as "spikiness" are not defined, but it's key to following the main theoretical result and the comparison to standard matrix completion.

c. why is there no RMSE table for the 3 large datasets? from Fig.2 it's unclear if RMSE of GRALS is equal/poorer than the other methods (it's clear that GRALS scales better)
Summary: The optimization problem considered in this paper -- graph-structed matrix factorization with partial observations -- appears to be novel and is likely to be of significant interest to the collaborative filtering community. The solution is based on weighted norm minimization (similar to work in Srebro and Salakhutdinov (2010)). The alternating least squares algorithm developed by the authors is convincingly shown to be much more efficient than SGD, and performs better than leading methods that include side information. However, I found the writing in key sections difficult to follow, and I haven't checked proofs.

Author Feedback
Author rebuttal: We thank the reviewers for their careful feedback and kind comments. We respond to the comments by each reviewer in detail below:

Reviewer_1:

Regarding our optimization innovations:
In addition to what was noted by the reviewer, another major contribution of our work is the efficient Hessian vector multiplication schemes (Algorithms 1 and 2) that enable the use of conjugate gradient methods to be applied in the collaborative filtering setting. The Hessian vector multiplication methods coupled with conjugate gradient makes GRALS highly efficient, much more so than just a direct application of conjugate gradient schemes.

Reviewer_2:

1. Regarding lack of comparison to the 3 suggested papers:
Salakhutdinov and Mnih, Probabilistic Matrix Factorization [NIPS 2008], Mackey et al., Divide-and-Conquer Matrix Factorization [NIPS 2011], Lee et al., Local Low-Rank Matrix Approximation [ICML 2013].

The SGD method we compare to in the paper was proposed in Zhou et al. 2012, which extends the work of Salakhutdinov and Minh to the kernel setting. Section 4.2 in Zhou et al shows that their method generalizes PMF, and hence we believed that a separate comparison to the latter was unnecessary.

The methods of Mackey et al, and Lee et al solve the standard matrix factorization problem, and do not take into account additional graph information among variables, and hence we did not compare to these methods in Figure 2. We will be happy to include RMSE results from these methods (and some others) in the final version of the paper.

2. Regarding the suggestion to also run the algorithm on Netflix or Yahoo Music datasets:

The Netflix dataset only includes the year of release as side information for movies, and no user information. While the Yahoo! music data does not have a graph over users, it does have features provided for the users, in much the same way as movielens, so that we could construct a knn graph from these features. We will be happy to do so, and provide a comparison in the final version of the paper.

3. We thank the reviewer for the suggestions regarding citations and axis labels, and will make the changes in the final version.

Reviewer_3:

1. Regarding the terse description of how the row/column graphs are generated from the movielens dataset:

We will expand upon our descriptions in the final version.

The dataset contains user features such as age (numeric), gender (binary), occupation, etc. which we map into a 22 dimensional feature vector per user. We then construct a 10-nearest neighbor graph using the euclidean distance metric. We do the same for the movies, except in this case we have an 18 dimensional feature vector per movie.

2. Regarding Sections 5 and 5.1 being difficult to follow, and terms not being defined:

We define the quantities \alpha and \beta in (14) and we mention in Section 5.1 that \alpha is the spikiness. As suggested by the reviewer, we will expand upon our descriptions, and add further explanations, in the final version.

3. Why is there no RMSE table for the 3 large datasets? from Fig.2 it's unclear if RMSE of GRALS is equal/poorer than the other methods (it's clear that GRALS scales better)

Figure 1 was devoted to an RMSE comparison, on the movielens dataset, among different statistical estimators, while Figure 2 was devoted to comparing the scalability of just those algorithms that take the graph into account, on some other larger datasets.

We will gladly add RMSE comparison over the datasets from Fig. 2 as well, to the appendix in the final version (for reasons of space). We also report the results below:

DATASET PMF GRALS SoREC SoREG RTSE
Epinions 1.173 1.162 1.144 1.232 1.278
Flixster 0.923 0.845 1.018 1.034 0.975

PMF : the method of Salakhutdinov and Minh, and SoREC, SoREG and RTSE are variants of methods proposed by Ma et al, 2009. We will add results for Douban as well in the final version.

Reviewer_4, Reviewer_6:

Thank you for your kind comments.